# Acceptance and Commitment Therapy (ACT) Improves Sleep Quality, Experiential Avoidance, and Emotion Regulation in Individuals with Insomnia—Results from a Randomized Interventional Study

**DOI:** 10.3390/life11020133

**Published:** 2021-02-09

**Authors:** Ali Zakiei, Habibolah Khazaie, Masoumeh Rostampour, Sakari Lemola, Maryam Esmaeili, Kenneth Dürsteler, Annette Beatrix Brühl, Dena Sadeghi-Bahmani, Serge Brand

**Affiliations:** 1Sleep Disorders Research Center, Kermanshah University of Medical Sciences, Kermanshah 6715847141, Iran; hakhazaie@kums.ac.ir (H.K.); m.rostampour@kums.ac.ir (M.R.); dena.sadeghibahmani@upk.ch (D.S.-B.); 2Department of Psychology, Bielefeld University, 33602 Bielefeld, Germany; sakari.lemola@uni-bielefeld.de; 3Department of Psychology, Faculty of Education, University of Isfahan, Isfahan 8174673441, Iran; m.esmaili@edu.ui.ac.ir; 4Psychiatric Clinics, Division of Substance Use Disorders Basel, University of Basel, 4055 Basel, Switzerland; Kenneth.Duersteler@upk.ch; 5Center for Addictive Disorders, Department of Psychiatry, Psychotherapy and Psychosomatics, Psychiatric Hospital, University of Zurich, 8008 Zürich, Switzerland; 6Center for Affective, Stress and Sleep Disorders (ZASS), Psychiatric University Hospital Basel, 4002 Basel, Switzerland; annette.bruehl@upk.ch; 7Department of Clinical Research, University of Basel, 4031 Basel, Switzerland; 8Departments of Physical Therapy, University of Alabama at Birmingham, Birmingham, AL 35294, USA; 9Substance Abuse Prevention Research Center, Kermanshah University of Medical Sciences, Kermanshah 6715847141, Iran; 10Division of Sport Science and Psychosocial Health, Department of Sport, Exercise and Health, University of Basel, 4052 Basel, Switzerland; 11School of Medicine, Tehran University of Medical Sciences, Tehran 1417466191, Iran

**Keywords:** insomnia, sleep quality, acceptance and commitment therapy, dysfunctional beliefs, sleep logs, experiential avoidance

## Abstract

Insomnia is a common problem in the general population. To treat insomnia, medication therapies and insomnia-related cognitive-behavioral interventions are often applied. The aim of the present study was to investigate the influence of acceptance and commitment therapy (ACT) on sleep quality, dysfunctional sleep beliefs and attitudes, experiential avoidance, and acceptance of sleep problems in individuals with insomnia, compared to a control condition. A total of 35 participants with diagnosed insomnia (mean age: 41.46 years old; 62.9% females) were randomly assigned to the ACT intervention (weekly group therapy for 60–70 min) or to the active control condition (weekly group meetings for 60–70 min without interventional and psychotherapeutic character). At baseline and after eight weeks (end of the study), and again 12 weeks later at follow-up, participants completed self-rating questionnaires on sleep quality, dysfunctional beliefs and attitudes about sleep, emotion regulation, and experiential avoidance. Furthermore, participants in the intervention condition kept a weekly sleep log for eight consecutive weeks (micro-analysis). Every morning, participants completed the daily sleep log, which consisted of items regarding subjective sleep duration, sleep quality, and the feeling of being restored. Sleep quality, dysfunctional beliefs and attitudes towards sleep, emotion regulation, and experiential avoidance improved over time, but only in the ACT condition compared to the control condition. Improvements remained stable until follow-up. Improvements in experiential avoidance were related to a favorable change in sleep and cognitive-emotional processing. Micro-analyses showed that improvements occurred within the first three weeks of treatment. The pattern of results suggests that ACT appeared to have improved experiential avoidance, which in turn improved both sleep quality and sleep-related cognitive-emotional processes at longer-term in adults with insomnia.

## 1. Introduction

A healthy lifestyle is associated with sufficient and restoring sleep [1]. However, in the general population, insomnia is common—the prevalence rates of insomnia in the general population may range from 9% to 15% [2], from 20% to 40% [3], or from 5% to 25% [4], depending on the underlying definition of and methodology to assess insomnia. Overall, about 30% of the general population reported having experienced insomnia at some point in their life [5]. Importantly, insomnia is related to higher risks of traffic accidents [6,7,8,9,10], more workplace absenteeism, impaired work performance [11,12,13], and to physiological issues such as diabetes [14] and cardiovascular diseases [15]. Further, insomnia predicted incidences of psychiatric disorders such as depression, anxiety, and alcohol abuse [16].

Following the International Classification of Sleep Disorders (ICSD-3) [17] chronic insomnia disorder involves the following symptoms: difficulty in initiating or maintaining sleep, early morning awakening with inability to return to sleep, and resistance to going to sleep. Daytime consequences include the feeling of not being restored, fatigue, excessive daytime sleepiness, and decreased attention, memory, and concentration. These impairments should occur at least three times per week for at least three months.

To treat insomnia, both pharmacological and non-pharmacological interventions are employed. Both the American Sleep Association (https://www.sleepassociation.org/sleep-treatments/cognitive-behavioral-therapy/ (accessed on 7 January 2021)) and the European Sleep Association [18] recommend the specific cognitive-behavioral therapy for insomnia (CBT-i) as first-line treatment. CBT-i can be delivered as group-therapy interventions [19,20] or as internet-delivered interventions [21,22]. Systematic reviews and meta-analyses confirmed that CBT-i improved sleep parameters, without adverse outcomes and side-effects [23]. Further non-pharmacological treatments involve exercise-based and psychotherapeutic interventions. Systematic reviews and meta-analyses showed that exercise-based interventions compared to no interventions led to improvements in some specific sleep parameters [24,25,26,27]. Likewise, mindfulness meditation improved subjective sleep among individuals with insomnia [28] and without insomnia [29]. For pharmacological treatment, compared to no treatment, placebo effects had a considerable impact on subjective improvements [30]. Consequently, alternative and psychotherapeutic interventions are considered. CBT interventions appeared to lead to medium [31] and clinically meaningful effect sizes [23], although risks of relapses were also reported [32]. Here, we considered acceptance and commitment therapy (ACT) as an intervention to treat insomnia.

Acceptance and commitment therapy (ACT) is a psychotherapeutic intervention based on the principles of cognitive-behavioral therapy (CBT) [33,34,35,36,37,38,39,40,41]. Briefly, key elements of ACT include (1) acceptance of all kind of emotions and thoughts as processes of the mind without acting out these emotions (e.g., sadness, anger, disappointments), without taking thoughts as ultimate truths (e.g., “I’m worthless!”; “I’m not worth being loved!”); or for short: “I do not avoid such emotions and cognitions, but I accept them as products of the mind.” (acceptance). The opposite of accepting unpleasant cognitions and emotions is “experiential avoidance.” Following others [42,43], experiential avoidance is the tendency to avoid unpleasant inner states—unpleasant feelings, thoughts, memories, images, impulses, and bodily perceptions are avoided. The disadvantage of such avoidance behavior is that this behavior needs cognitive-emotional resources. This leads to two consequences: First, cognitive-emotional resources are continuously entangled with unpleasantness, and second, cognitive-emotional resources are not available to pursue core values and to pursue what is more important in life. Accepting unpleasant feelings and thoughts, that is, giving-up to avoid inner experiences (experiential avoidance) means accepting unpleasant feelings and thoughts as transient processes of the mind without acting out these emotions and without taking such feelings and thoughts as ultimate truths; (2) identifying core values, i.e., what is really important in one’s life, being committed to these values and related behavior?—“The more I’m aware of my core values, the more my behavior is oriented towards the achievement of these values.”; (3) “focusing my behavior to achieve values equals to be committed both to the values and related behavior” (commitment); and (4) “ACT is both the acronym of the intervention and the chief focus on doing, being active, and behaving actively to move toward my core values.” Authors investigating the underlying psychological mechanisms of ACT underscore the importance of acceptance and not avoiding experiences anymore (experiential avoidance), psychological flexibility, and staying committed to the core values and their actions to turn values into action [33,34,35,36,37,38,39,40,41].

Results from meta-analyses and systematic reviews showed that ACT is always a valuable psychotherapeutic intervention for major depressive disorders [44], anxiety disorder [45], posttraumatic stress disorders [46], and insomnia [47] when compared to no-treatment conditions. In a single case study, ACT impacted favorably on sleep quality [32]. In two previous pilot studies ACT interventions for eight weeks improved subjective sleep quality among four and 15 individuals with insomnia [48,49]. An ACT intervention improved sleep patterns in adults with insomnia (n = 43), compared to a wait-list condition (n = 40) [50]. ACT-interventions improved subjective sleep among ten individuals with insomnia poorly responding to CBT-interventions before [51]. Importantly, improvements appeared to remain stable over a six-month [50] and nine-month period [52]. To explain the favorable impact of ACT on insomnia, aspects of mindfulness [53] and decreased experiential avoidance [54] appeared to decrease psychological arousal.

The current study expanded upon previous research in four ways—First, we introduced an active control condition, i.e., participants in the control condition gathered in small groups once a week in the study center. In doing so, we were able to partial out possible improvements in the ACT condition due to a mere social context. Second, participants in the ACT condition completed a daily log for eight consecutive weeks. In doing so, we were able to observe, if and at which time points ACT improved dimensions of sleep patterns and cognitive-emotional processes. Third, the follow-up occurred 12 weeks after the end of the intervention, which allowed an estimate, if and to what extent ACT interventions could impact cognitive-emotional processes in the longer term. Fourth, we associated experiential avoidance, a key concept to explain ACT-related improvements, with dimensions of sleep quality, emotion regulation, dysfunctional beliefs and attitudes about sleep, and acceptance of sleep problems. To this end, 35 adults with insomnia were randomly assigned to the ACT intervention and to the active control condition, and dimensions of sleep quality and cognitive-emotional processes were self-assessed at baseline, eight weeks later at the end of the study, and again 12 weeks later at follow-up.

The following three hypotheses and one research question were formulated: first, following others [32,47,48,49,50,51], we expected improvements on subjective sleep and cognitive-emotional processes in the intervention-/ACT-condition, compared to an active control condition. Second, following others [34,40,54,55,56] we assumed that improvements in experiential avoidance would be associated with favorable changes in sleep quality and cognitive-emotional processes. Third, following Lappalainen et al. [50] and Daly-Eichenhardt et al. [52], we expected that improvements would remain stable until follow-up 12 weeks after study completion. We treated the study question as exploratory, asking at which time point (week) of the intervention improvements in sleep patterns could be observed, compared to baseline. Answering these research questions is of clinical importance because in clinical practice it may complement the CBT-i, which, although generally effective, may not be effective in all individuals with insomnia. Results are also of practical importance because ACT may particularly improve experiential avoidance, which could be associated with a broad range of further favorable cognitive-emotional processes.

## 2. Methods

### 2.1. Study Procedure

Individuals with clinically diagnosed insomnia from the Sleep Disorders Research Center of the Kermanshah University of Medical Sciences (Kermanshah, Iran) were approached between January 2019 and August 2020 to participate in the present interventional study. Eligible participants were fully informed about the study and the secure and anonymous data handling. Thereafter, they signed the written informed consent and they were randomly assigned either to the intervention condition (acceptance and commitment therapy; ACT; see Section 2.6) or to an active control condition (see Section 2.7). At the beginning, eight weeks later at the end of the study, and again 12 weeks later at follow-up participants completed a booklet of questionnaires covering sociodemographic and sleep-related information (see Section 2.5). Further, participants in the ACT condition were keeping a weekly sleep log for eight consecutive weeks (see Section 2.5.3). An experienced and certified psychotherapist on ACT performed the group interventions of ACT once a week for 60–70 min. Meetings of participants in the active control condition were identical as regards frequency and duration as meetings in the ACT condition (see Section 2.7). The Kermanshah University of Medical Science Ethics Committee (IR.KUMS.REC.1396.478) approved the study, which was performed in accordance with the seventh and last revision [57] of the Declaration of Helsinki. Two previous feasibility and pilot studies at a small scale with no control condition [58,59] pawed the grounds for the present study.

### 2.2. Participants

A total of 45 individuals with insomnia were approached. Inclusion criteria were (1) suffering from insomnia, based on self-reports and on a structured clinical interview for DSM-5 psychiatric disorders [60]; (2) age between 18 years old and 60 years old; (3) willing and able to comply with the study conditions; and (4) signed written informed consent. Exclusion criteria were (1) psychiatric disorders such as major depressive disorders, bipolar disorders, cognitive impairments, or substance use disorder with insomnia as a consequence of the main psychiatric disorder; to exclude psychiatric disorders the same structured clinical interview for DSM-5 psychiatric disorders [60] was employed; (2) insomnia due to shift work or family issues such as small children, children with mental and/or physical impairments that would explain the emergence and maintenance of insomnia; (3) restless legs syndrome or sleep-disordered breathing, as assessed via the clinical interview; (4) intake of medications with sleep- and mood-altering effects; and (5) undergoing interventions such as psychotherapy, including CBTi, relaxation techniques or physical exercising to improve sleep problems.

Of the 45 individuals approached, three did not meet the inclusion criteria, and two declined to participate. Hence, 40 individuals (88.89%) were randomly assigned to either the ACT intervention or to the active control condition (see Figure 1). A total of 35 participants accomplished the study from baseline to follow-up. All outcome analyses were performed on completer data.

### 2.3. Sample Size Calculation

The sample size calculation was performed with G*Power^®^ [61]. Based on previous results of two pilot studies [58,59], the following parameters were defined: effect size: partial eta-squared: 0.08; Cohen’s f for ANOVAs: 0.29; alpha error probability: 0.05; power: 0.95; number of groups: 2; and number of measurements: 3. These parameters yield a total sample size of 32. However, to allow for dropouts, the sample size was set at 40 participants.

### 2.4. Randomization

As described elsewhere [62,63,64], randomization was accomplished using the software randomization.com to create a list to assign 40 participants randomly to one of the two study conditions. Thereafter, a psychologist not otherwise involved in the study managed the assignments.

### 2.5. Measures

#### 2.5.1. Experiential Avoidance

The Experiential Avoidance Questionnaire (EAQ) [55] was used, which consists of 10 items to assess experiential avoidance. Typical items include “I am afraid of my feelings.” and “My thoughts and feelings mess up my life.” Answers are given on 7-point Likert scales ranging from 1 (= never) to 7 (= always). Higher sum scores reflect a higher degree to accept also unpleasant experiences (emotions, cognitions), and accordingly, a lower degree of avoidance (Cronbach’s alpha = 0.85).

#### 2.5.2. Subjective Sleep: Pittsburgh Sleep Quality Index (PSQI)

As described elsewhere [62], the Pittsburgh Sleep Quality Index PSQI [65] is a self-report scale completed in five minutes. It consists of 19 items and contains seven subscales (subjective sleep quality, sleep latency, sleep duration, sleep efficiency, sleep disturbance, sleeping medication, and daytime dysfunction), each weighted equally on a scale from 0 to 3, with higher scores indicating poorer sleep quality. The seven components are then summed to obtain an overall PSQI score, ranging from 0 (good sleep quality) to 21 (poor sleep quality). Total scores of ≥5 reflect poor sleep, associated with considerable sleep complaints. The Farsi versions of PSQI have been validated for adults [66], older adults [67], and adolescents [68] (Cronbach’s alpha = 0.85).

#### 2.5.3. Sleep Log

Participants in the intervention condition completed every morning a sleep log for eight consecutive weeks. Items were taken from a widely used manual on self-treatment of insomnia [69]. The items were (1) time to go to bed—answer in h and min (e.g., 22.30 h); (2) sleep onset duration—answer in min; (3) the number of awakenings after sleep onset; (4) duration of time awake after sleep onset; (5) wake-up time; (6) sleep quality—answer based on a 9-point Likert scale: 1 = very poor sleep quality; 9 = very good sleep quality; (7) feeling of being restored—answer based on a 9-point Likert scale: 1 = not at all restored; 9 = completely restored. Total sleep time (in h and min) was calculated based on the answers 1 to 5. Data from five weekdays were merged to overall scores for one week.

#### 2.5.4. Dysfunctional Beliefs and Attitudes about Sleep (DBAS)

To assess dysfunctional pre-sleep beliefs and attitudes, participants completed the Dysfunctional Beliefs and Attitudes about Sleep (DBAS) questionnaire [70]. The questionnaire consists of 10 items. Typical items include “I need 8 h of sleep to feel refreshed and function well during the day.” and “When I don’t get proper amount of sleep on a given night, I need to catch up on the next day by napping or on the next night by sleeping longer.” Answers are given on 10-point Likert scales, ranging from 1 (= not at all true) to 10 (= completely true), with higher sum scores reflecting more pronounced dysfunctional beliefs and attitudes about sleep (Cronbach’s alpha = 0.79).

#### 2.5.5. Sleep Problem Acceptance Questionnaire (SPAQ)

To assess the acceptance to suffer from sleep problems, participants completed the Sleep Problem Acceptance Questionnaire (SPAQ) [71]. The questionnaire consists of eight items. Typical items are “Although things have changed, I am living a normal life despite my sleeping problems.” and “I live a full life even though I have sleeping problems.” Answers are given on 7-point Likert scales, ranging from 1 (= not at all true) to 10 (= completely true), with higher sum scores reflecting a higher acceptance to suffer from sleep problems (Cronbach’s alpha = 0.85).

#### 2.5.6. Difficulties in Emotion Regulation Scale (DERS)

To assess difficulties in emotion regulation, participants completed the Difficulties in Emotion Regulation Scale (DERS) [72]. The questionnaire consists of 36 items. Typical items include “I know my feelings.” and “I pay attention to how I feel.” Answers are given on 5-point Likert scales, ranging from 1 (= not at all true) to 5 (= completely true), In the present study, we considered the overall sum score—a higher sum score reflects a higher degree to regulate their emotions. (Cronbach’s alpha = 0.93).

### 2.6. Intervention: Acceptance and Commitment Therapy (ACT)

The ACT intervention consisted of eight weekly group sessions lasting for 60–70 min. A group consisted of eight to nine participants. For the content and the structure of the sessions, we followed well-established therapy manuals [42,43,56,73,74]. A clinical psychologist and certified psychotherapist in ACT (AZ) was responsible for the ACT sessions.

Table 1 provides the content of the eight sessions in ACT.

### 2.7. Active Control Condition

As described extensively elsewhere [75,76,77,78,79,80], participants in the control condition met in groups with a staff member and social worker once a week for eight consecutive weeks for about 60–70 min. They had group discussions on daily activities and daily problems. The control condition could not be considered as a bona fide intervention [75,76,77,78,79,80], given that the sessions with the staff member and social worker excluded treatment elements that were truly intended to be therapeutic [81]. Participants were just encouraged to exchange daily life experiences. Blood pressure was also checked once per week. The active control condition was not intended to be an active therapy but to control for placebo effects in the intervention condition.

### 2.8. Statistical Analysis

A series of *t*-tests and *X*^2^-tests was performed to compare sociodemographic and sleep-related data at baseline between participants in the intervention and in the control condition. At baseline, and compared to the active control condition, participants in the intervention-/ACT-condition reported subjectively lower sleep complaints (statistically significantly lower PSQI scores; see Table 2). Given this evidence, PSQI at baseline was introduced as the covariate. Accordingly, a series of ANCOVAs for repeated measures was performed with the factors time (baseline, study end, follow-up), group (intervention versus control condition) and the time by group interaction; subjective sleep quality, dysfunctional beliefs and attitudes about sleep, sleep problems acceptance, difficulties in emotion regulation, and experiential avoidance were dependent variables. In case of deviation from sphericity, ANCOVAs were computed using Greenhouse–Geisser corrected degrees of freedom, though the original degrees of freedom are reported with the relevant Greenhouse–Geisser epsilon value (ε). Post-hoc tests were performed with Bonferroni–Holm corrections for *p*-values. For *t*-tests, Cohen’s d effect sizes are reported. For F-tests, partial eta-squared’s ($\eta_{p}^{2}$) effect sizes are reported. Cut-off values for Cohen’s d’s were d < 0.19 = trivial effect size (T); 0.20 < d < 0.49 = small effect size (S); 0.50 < d < 0.79 = medium effect size (M); d > 0.80 = large effect size (L). Cut-off values for partial eta-squared were: $\eta_{p}^{2}$ < 0.019 = trivial effect size (T); $0.02<\eta_{p}^{2}$ < 0.059 = small effect size (S); 0.06 < $\eta_{p}^{2}$ < 0.139 = medium effect size (M); $\eta_{p}^{2}$ > 0.14 = large effect size (L).

For micro-analysis of weekly change in total sleep time, sleep quality, and the feeling of being restored, we followed Becker [82] and compared standard mean changes with Cohen’s d’s effect sizes.

With a series of Pearson’s correlations associations between improvements in experiential avoidance (that is, less avoidance of unpleasant experiences at the end of the study after the intervention) and sleep quality, dysfunctional beliefs and attitudes about sleep, sleep problems acceptance, and difficulties in emotion regulation were calculated.

All statistical computations were performed with SPSS^®^ 25.0 (IBM Corporation, Armonk, NY, USA) for Apple Mac^®^.

## 3. Results

### 3.1. Sociodemographic and Sleep-Related Information at Baseline

Table 2 provides the descriptive and statistical overview of sociodemographic and sleep-related information at baseline between the participants in the intervention and control condition. Statistical indices are not repeated in the text anymore.

Compared to participants in the control condition, participants in the intervention condition reported higher educational levels, a higher acceptance of sleep problems, and a lower subjective sleep disturbance (lower PSQI score). Given this evidence, the PSQI score at baseline was introduced as the covariate. There were no descriptive and statistically significant mean differences for age, sex distribution, current job position, civil status, subjective sleep quality, dysfunctional beliefs and attitudes about sleep, difficulties in emotion regulation, and experiential avoidance.

### 3.2. Changes in Subjective Sleep Quality, Dysfunctional Beliefs and Attitudes about Sleep, Sleep Problems Acceptance, Difficulties in Emotion Regulation, and Experiential Avoidance over Time between and within the Intervention and Control Conditions

Table 3 and Table 4 provide the descriptive and inferential statistical overview of subjective sleep quality, dysfunctional beliefs and attitudes about sleep, sleep problems acceptance, difficulties in emotion regulation, and experiential avoidance over time between and within the intervention and control conditions and always controlling for PSQI baseline scores.

Experiential avoidance and dysfunctional beliefs and attitudes about sleep decreased over time (the significant factor was time). Subjective sleep quality, sleep problems acceptance, difficulties in emotion regulation did not statistically change over time. Significant time by group interactions showed that experiential avoidance, dysfunctional beliefs and attitudes about sleep, subjective sleep quality, sleep problems acceptance, and difficulties in emotion regulation decreased over time in the intervention-/ACT condition but not in the active control condition. Post-hoc comparisons with Bonferroni–Holm corrections for *p*-values showed that compared to the control condition, participants in the intervention condition reported more favorable scores in all dimensions (the significant factor was group). Importantly, improvements remained stable 12 weeks after the end of the intervention. Figure 2 illustrates mean changes over time between and within groups for experiential avoidance.

### 3.3. Micro-Analysis; Weekly Changes in Total Sleep Time, Sleep Quality, and the Feeling of Being Restored in the Intervention Group

Table 5 provides the effect sizes of weekly comparisons of sleep duration, sleep quality, and the feeling of being restored, and Figure 3, Figure 4 and Figure 5 show the graphical solutions.

For total sleep time, significant improvements (medium and large effect sizes) were observed from baseline to week 2, from week 2 to week 3, and again from week 6 to week 7 (medium effect size).

For subjective sleep quality, there were no significant improvements from week to week (trivial to small effect sizes). Subjective sleep quality, however, improved from baseline to weeks 3 to 8 (always large effect sizes).

For the feeling of being restored, significant improvements were observed from baseline to week 2 and from week 2 to week 3 (medium effect sizes). From week 3, week-to-week comparisons yielded no significant improvements (trivial to small effect sizes).

Overall, significant improvements were found within the first three weeks after the beginning of the intervention.

### 3.4. Changes in Experiential Avoidance and Associations with Sleep Quality, Dysfunctional Beliefs and Attitudes about Sleep, Sleep Problems Acceptance, and Difficulties in Emotion Regulation at the End of the Study in the Intervention Condition

To improve sleep, the psychotherapeutic intervention was ACT; to operationalize ACT, we employed the Experiential Avoidance Questionnaire (AAQ). Improvements in avoiding unpleasant experiences (that is, lower scores in experiential avoidance) should be associated with improvements in sleep quality, dysfunctional beliefs and attitudes about sleep, sleep problems acceptance, and difficulties in emotion regulation at the end of the study.

A higher change in experiential avoidance over time, that is, a lower attitude to avoid unpleasant experiences, was associated with lower dysfunctional beliefs and attitudes about sleep (DBAS, r = −0.45, *p* < 0.05), fewer difficulties in emotion regulation (DERS; r = −0.14, ns), a higher degree to accept sleep problems (SPAQ; r = −0.14, ns), and a higher sleep quality (PSQI; r = −0.59, *p* < 0.05).

## 4. Discussion

The key findings of the present study were that among a sample of individuals with insomnia an eight-week acceptance and commitment therapy (ACT) group intervention improved experiential avoidance, sleep quality, dysfunctional beliefs and attitudes about sleep, sleep problems acceptance, and difficulties in emotion regulation, as compared to an active control condition. Improvements were above all observed within the first three weeks of intervention. Further, improvements remained stable for further 12 weeks until follow-up. The present pattern of results was comparable and fit well with those reported recently in a systematic review on the influence of ACT on insomnia and sleep quality [47]. However, the present results expand upon previous research in that ACT interventions affected the sleep quality and the underlying dysfunctional cognitive-emotional processes. As such, it is conceivable that such improvements in cognitive-emotional processes may also explain the therapeutic effect in the long-term.

Three hypotheses and one research question were formulated and each of them is considered as follows.

With the first hypothesis, we assumed that subjective sleep and cognitive-emotional processes would improve over time in the ACT condition, compared to the active control condition, and data did support this. Accordingly, the present data are in accordance with previous results [32,47,50]. However, the present results expand upon previous studies in three ways—(1) improvements at the end of the interventions remained stable for 12 additional weeks until follow-up; (2) unlike previous studies [32,47,50,58,59], we employed an active control condition to partial out possible treatment effects due to the social context of group therapy; and (3) the micro-analysis allowed a fine-grained measurement and assessment of sleep-related improvements.

With the second hypothesis, we expected that improvements in experiential avoidance would be associated with favorable changes in sleep quality and cognitive-emotional processes, and data did confirm this. Accordingly, the present data both replicated previous findings [34,40,54,55,56] and expanded on previous results in three ways—(1) we employed a questionnaire to assess experiential avoidance, which by definition, is one of the key concepts of ACT, namely, accepting prevalently unpleasant feelings and dysfunctional thoughts, or the other way around, not avoiding such unpleasant feelings and dysfunctional thoughts anymore. Table 3 and Table 4 display that experiential avoidance improved over time in the ACT condition but not in the active control condition, thereby suggesting that improvements in experiential avoidance were associated with ACT-related interventions; 2) following the transdiagnostic approach [83,84,85,86,87,88], it appears plausible that improvements in sleep quality are associated with favorable changes in the underling cognitive-emotional processes; and (3) improvements in sleep quality and quantity were above all observed within the first three weeks of treatment.

With the third hypothesis, we assumed that ACT-related improvements on sleep remained stable also for 12 weeks later at follow-up, and again data did confirm this. Accordingly, the present results both replicated [50,52] and expanded upon previous studies in the following two ways—(1) improvements in the ACT-condition were always compared to an active control condition, and (2) we measured if and to what extent experiential avoidance remained stable over time as a proxy of the learning process and interiorization of the acquired ACT-related skills during the intervention.

Lastly, we took the study question as exploratory, asking at which time point (week) of the intervention improvements in sleep patterns could be observed. As shown in Table 5 and in Figure 3, Figure 4 and Figure 5, improvements in sleep duration, sleep quality, and the feelings of being restored occurred within the first three weeks of intervention, followed by further descriptive but statistically nonsignificant improvements. To our knowledge, no such data have been found before. A closer look at the specific interventions employed within the first four weeks reveals (see Table 1) that participants acquired key concepts of ACT—accepting unpleasant feelings and thoughts as simple products of the mind, to which we are not obliged to react (acceptance, cognitive diffusion); thinking about personal core values and how to act and move to get closer to such core values, while staying committed to them (commitment); and techniques of mindfulness. In contrast, the content of the sessions in weeks five to eight might be more considered as consolidation of what has been acquired before [42]. As such, it appears plausible that the learning curve changed from a rapid and steep improvement to a rather asymptotic process.

Despite the novelty of the results, the following limitations warrant against overgeneralization. First, the sample size was small, though, we focused on effect size calculations, which by definition are not sensitive to sample sizes. Second, participants were prevalently women (62.9%); given this fact, it is conceivable that a balanced sex ratio would have yielded a modified pattern of results. Third, we assessed a highly selected group of adults with insomnia self-reporting a sleep duration of four hours per night. In contrast, clinical and everyday experiences show that adults with insomnia usually report further psychological issues such as symptoms of depression, anxiety, and rumination. As such, it is conceivable that the present sample does not appropriately reflect every day and clinical reality. Fourth, except for the diagnosis, we fully relied on self-reports; therefore, experts’ ratings might have strengthened the validity of the results. Fifth, similarly, psychophysiological indices such as cortisol concentrations as a proxy of psychophysiological arousal could have informed us if subjective sleep improvements were accompanied by psychophysiological changes. Sixth, by definition, individuals with insomnia claim an insufficiently short sleep duration (see Table 5), while objective sleep-EEG assessments do not confirm such claims. Given this background, it would have been important to see if and to what extent subjective and objective sleep duration do match or not, and if ACT interventions would have yielded a more accurate sleep duration estimation when compared to objective measurements; simply put, it would have been interesting to see if improvements in experiential avoidance were associated with a more accurate objective sleep duration estimation. Seventh, while we claimed that improvements in experiential avoidance as the key factor of ACT was the driver of improved sleep, most of the change in sleep occurred within the first three weeks of treatment; given this fact, the change in experiential avoidance should have had to occur even earlier in order to establish an adequate timeline between outcome and presumed change mechanism. Eighth, it would have been interesting to know if participants’ family members observed participants’ behavioral and sleep-related changes. Future studies might introduce the so-called one-for-many procedure, where family members and peers are asked to evaluate the behavior of study participants [89].

## 5. Conclusions

The results of the present randomized intervention study suggested that ACT had the potential to favorably impact subjective sleep and cognitive-emotional processes related to insomnia among adults with insomnia. Improvements were observable in the first three to four weeks; improvements remained stable for 12 additional weeks after the end of the study.

## Figures and Tables

**Figure 1 life-11-00133-f001:**
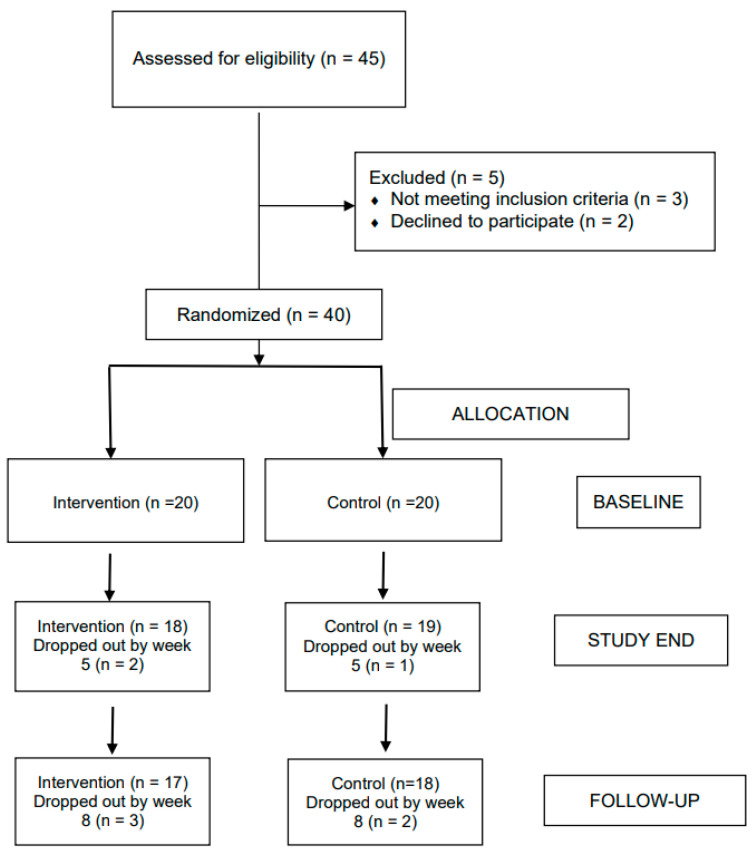
CONSORT diagram showing the flow of participants through each stage.

**Figure 2 life-11-00133-f002:**
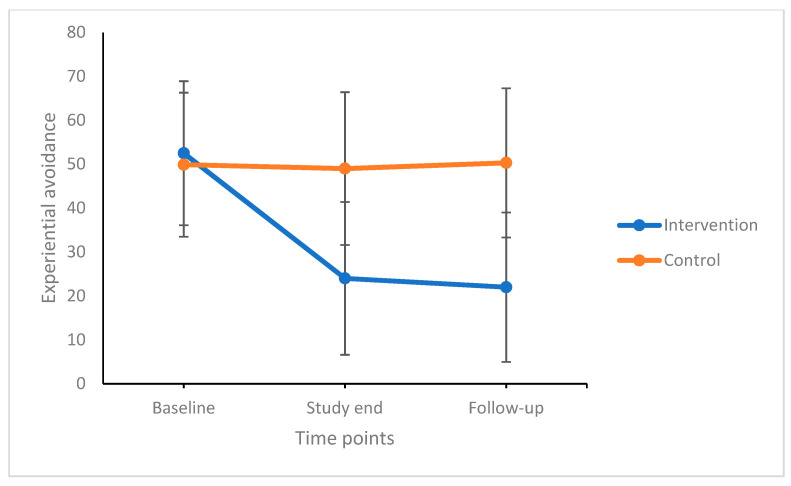
Experiential avoidance decreased over time, but more so in the intervention condition compared to the control condition. Improvements at the end of the study remained stable until follow-up 12 weeks later. Points are means and bars are standard deviations.

**Figure 3 life-11-00133-f003:**
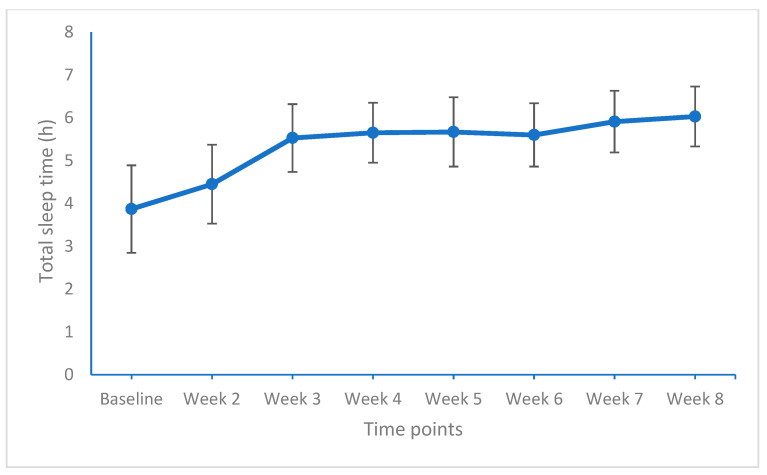
Total sleep time (h) from baseline to week 8 (end of the study). Points are means and bars are standard deviations.

**Figure 4 life-11-00133-f004:**
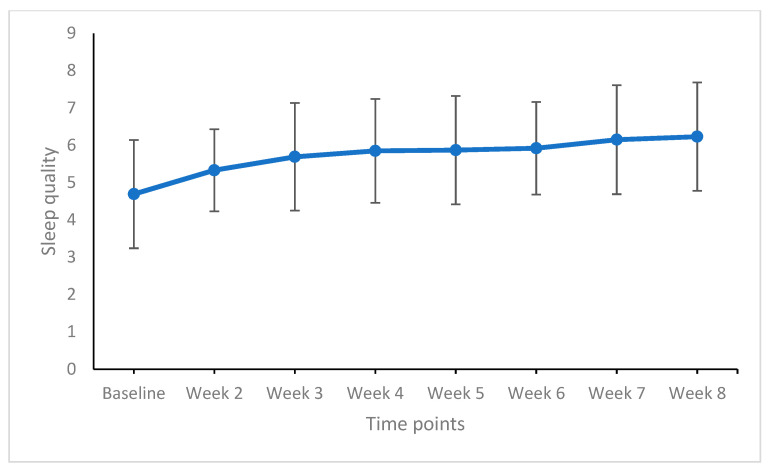
Sleep quality from baseline to week 8 (end of the study). Points are means and bars are standard deviations. Higher means reflect a higher sleep quality.

**Figure 5 life-11-00133-f005:**
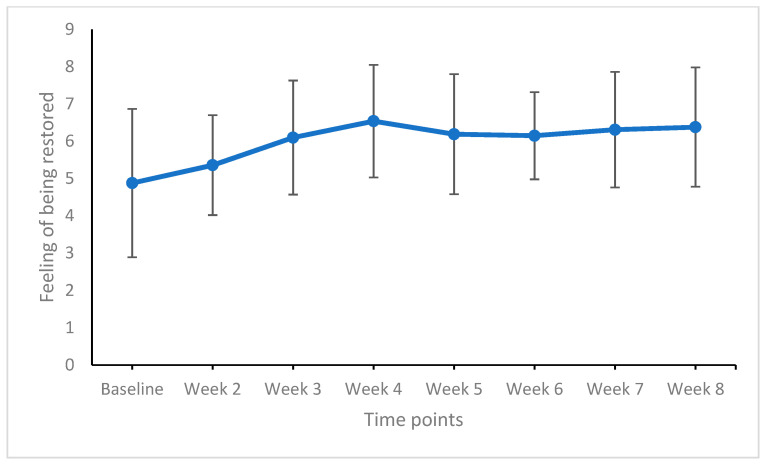
Feeling of being restored from baseline to week 8 (end of the study). Points are means and bars are standard deviations. Higher means reflect a higher feeling of being restored.

**Table 1 life-11-00133-t001:** Overview of the content of the eight acceptance and commitment therapy (ACT) sessions.

Session	Specification
Session One	Assessment of current sleep patterns and sleep quality. Individual explanations of insomnia (health beliefs models); Explanation of the key concepts of ACT (the metaphor of dirt and glasses; metaphor of two mountains); Gathering unpleasant feelings and thoughts; explanation that feelings and thoughts are just transient products of the mind; the key concept of acceptance and experience avoidance; exploring individual values; how to become what I want to be (metaphor of the grave stone; what is written on the grave stone about my life?) Achieving goals means action; ACT as acronym and emphasis of action and behavior Establishing a therapeutic relationship; creative despair (metaphor of the hole), mindfulness (breathing exercise), homework
Session Two	Discussing homework. Repetition of key concepts of ACT; concept of creative hopelessness; exercising mindfulness How to cope with pre-sleep dysfunctional thoughts and feelings Accepting instead of fighting for control: exercise with the rope: pulling or leaving? Exercise to fight with monsters. Are monsters rather enemies or buddies? Homework: Paying attention to experiences, thoughts, and feelings; recording them
Sessions Threeand Four	Discussing homework; listening to participants’ feelings, thoughts, and experiences; demask the power of thoughts and feelings: singing and distorting feelings and thoughts to make them ridiculous. Repetition: “Control is the problem, and acceptance is the solution”, how to measure experiential avoidance, and how to promote acceptance and commitment; how to deal with barriers Barriers: passenger in a bus; how to cope with a disobedient child Keeping values in mind; reducing the gap between current state and goals Staying committed to the action Willingness and acceptance (metaphor of the scale), homework
Sessions Fiveand Six	Explanation about self-context and dissonance, reviewing sleep pattern, and exercising mindfulness, practice of naming thoughts, introducing and explaining the concepts of self-context and cognitive dissonance (metaphor of chessboard and lion exercise) Exercising mindfulness (thoughts and emotions on paper), homework
Sessions Sevenand Eight	Discussing homework; repeating key concepts of ACT; emphasis on behavior and action; emphasis to distinguishing what I can control (=> action; commitment to actions), and what is out of my control (=> accept) Reviewing sleep pattern and exercising mindfulness How to implement ACT in everyday life (metaphor of dialysis)

**Table 2 life-11-00133-t002:** Descriptive and inferential statistical overview of sociodemographic, sleep, and psychological functioning at baseline, separately for individuals with insomnia in the intervention (*n* = 17) and the control (*n* = 18) conditions.

	Groups		Statistics
	Intervention	Control	
N	17	18	
Sex (female/male)	10/7	12/6	X^2^(N = 35, df = 1) = 0.23
Civil status (single/married)	5/12	6/12	X^2^(N = 35, df = 1) = 0.06
Highest education (middle school/diploma/higher education)	2/10/5	8/4/6	X^2^(N = 35, df = 1) = 6.24 *
Current job position (unemployed/employed)	12/5	8/10	X^2^(N = 35, df = 1) = 2.58
	M (SD)	M (SD)	
Age (years)	41.45 (8.66)	41.47 (7.48)	t(33) = 0.01
Weight	69.89 (11.37)	74.35 (11.32)	t(33) = 1.16
Pittsburgh Sleep Quality Index (PSQI)	15.76 (2.13)	18.00 (3.14)	t(33) = 2.44 * d = 0.83 (L)
Dysfunctional Beliefs and Attitudes about Sleep (DBAS)	90.82 (10.44)	84.78 (12.62)	t(33) = 1.54 d = 0.52 (M)
Sleep Problem Acceptance (SPAQ)	17.76 (6.75)	11.22 (6.52)	t(33) = 2.92 ** d = 0.98 (L)
Experiential Avoidance (EA)	52.70 (8.97)	49.89 (16.43)	t(33) = 0.62 d = 0.21 (S)
Difficulties in emotion regulation scale (DERS)	108.88 (28.67)	126.44 (27.87)	t(33) = 1.84 d = 0.62 (M)

Notes: * *p* < 0.05; ** *p* < 0.01; S = small effect size; M = medium effect size; L = large effect size.

**Table 3 life-11-00133-t003:** Descriptive statistical overview of sleep-related indices and psychological functioning at baseline, at the end of the eight-week study and at follow-up after 12 weeks, separately for individuals with insomnia in the intervention (*n* = 17) and the control (*n* = 18) conditions.

	Baseline		Study End		Follow-Up	
	Intervention	Control	Intervention	Control	Intervention	Control
N	17	18	17	18	17	18
	M (SD)	M (SD)	M (SD)	M (SD)	M (SD)	M (SD)
Pittsburgh Sleep Quality Index (PSQI)	15.76 (2.13)	18.00 (3.14)	4.77 (2.54)	12.89 (5.26)	4.06 (1.75)	12.50 (4.58)
Dysfunctional Beliefs and Attitudes about Sleep (DBAS)	90.82 (10.44)	84.78 (12.62)	31.76 (16.002)	81.88 (13.32)	31.29 (14.69)	81.05 (13.59)
Sleep Problem Acceptance (SPAQ)	17.76 (6.75)	11.22 (6.52)	35.71 (7.39)	13 (7.93)	37.89 (8.56)	14.06 (7.53)
Experiential Avoidance (EA)	52.53 (8.71)	49.89 (16.43)	24.06 (8.81)	49.78 (17.36)	22.29 (7.72)	50.28 (17.03)
Difficulties in emotion regulation scale (DERS)	108.88 (28.67)	126.44 (27.87)	60.47 (13.26)	126.22 (27.11)	62.29 (14.68)	128.94 (25.42)

**Table 4 life-11-00133-t004:** Inferential statistical indices of sleep-related indices and psychological functioning, always controlling for PSQI scores at baseline.

	Inferential Statistics						
	Time		Group		Time × Group Interaction		Greenhouse-Geisser ε
	F	$\eta_{p}^{2}$ (ES)	F	$\eta_{p}^{2}$ (ES)	F	$\eta_{p}^{2}$ (ES)	
Sleep quality (PSQI)	F(2,64) = 0.843	0.026 (S)	F(1,32) = 29.94 ***	0.484 (L)	F(2,64) = 27.01 ***	0.458 (L)	0.596
Dysfunctional Beliefs and Attitudes about Sleep (DBAS)	F(2,64) = 5.36 *	0.143 (L)	F(1,32) = 39.40 ***	0.552 (L)	F(2,64) = 207.82 ***	0.867 (L)	0.529
Sleep Problem Acceptance (SPAQ)	F(2,64) = 0.092	0.003 (T)	F(1,32) = 40.69 ***	0.560 (L)	F(2,64) = 112.32 ***	0.778 (L)	0.742
Experiential Avoidance (EA)	F(2,64) = 2.31	0.832 (L)	F(1,32) = 8.36 **	0.207 (L)	F(2,64) = 141.89 ***	0.816 (L)	0.666
Difficulties in emotion regulation scale (DERS)	F(2,64) = 1.52	0.045 (S)	F(1,32) = 34.06 ***	0.516 (L)	F(2,64) = 34.92 ***	0.522 (L)	0.552

Notes: * *p* < 0.05; ** *p* < 0.01; *** *p* < 0.001; (T) = trivial effect size; (S) = small effect size; (M) = medium effect size; (L) = large effect size.

**Table 5 life-11-00133-t005:** Descriptive and inferential statistical overview of sleep diary data over all eight weeks in the intervention condition (*n* = 17).

		Statistics							
		**F $\eta_{p}^{2}$ ε**	**Week-by-Week Comparisons (Cohen’s d)**						
Total sleep duration (h)	M (SD)	F(7,112) 39.04 *** 0.709	BL-W2	BL-W3	BL-W4	BL-W5	BL-W6	BL-W7	BL-W8
Baseline	3.87 (1.02)		0.59 (M)	1.02 (L)	1.95 (L)	1.87 (L)	1.95 (L)	2.31 (L)	2.46 (L)
Week 2	4.45 (0.92)		-	W2-W3	W2-W4	W2-W5	W2-W6	W2-W7	W2-W8
			-	1.25 (L)	1.45 (L)	1.38 (L)	1.37 (L)	1.76 (L)	1.92 (L)
Week 3	5.53 (0.79)		-	-	W3-W4	W3-W5	W3-W6	W3-W7	W3-W8
			-	-	0.12 (T)	0.16 (T)	0.09 (T)	0.50 (M)	0.88 (L)
Week 4	5.65 (0.70)		-	-	-	W4-W5	W4-W6	W4-W7	W4-W8
			-	-	-	0.01 (T)	0.07 (T)	0.36 (S)	0.52 (M)
Week 5	5.67 (0.81)		-	-	-	-	W5-W6	W5-W7	W5-W8
			-	-	-	-	0.07 (T)	0.34 (S)	0.49 (S)
Week 6	5.60 (0.74)		-	-	-	-	-	W6-W7	W6-W8
			-	-	-	-	-	0.42 (M)	0.58 (M)
Week 7	5.91 (0.72)		-	-	-	-	-	-	W7-W8
			-	-	-	-	-	-	0.15 (T)
Week 8	6.03 (0.70)		-	-	-	-	-	-	-
		**F $\eta_{p}^{2}$ ε**	**Week-by-Week Comparisons (Cohen’s d)**						
Subjective sleep quality	M (SD)	F(7, 112) 10.82 *** 0.403 0.241	BL-W2	BL-W3	BL-W4	BL-W5	BL-W6	BL-W7	BL-W8
Baseline	4.69 (1.45)		0.49 (S)	0.82 (L)	0.82 (L)	0.81 (L)	0.91 (L)	0.99 (L)	1.04 (L)
Week 2	5.34 (1.10)		-	W2-W3	W2-W4	W2-W5	W2-W6	W2-W7	W2-W8
			-	0.43 (S)	0.41 (S)	0.42 (S)	0.50 (M)	0.62 (M)	0.69 (M)
Week 3	5.89 (1.44)		-	-	W3-W4	W3-W5	W3-W6	W3-W7	W3-W8
			-	-	0.03 (T)	0.01 (T)	0.02 (T)	0.51 (M)	0.24 (M)
Week 4	5.85 (1.39)		-	-	-	W4-W5	W4-W6	W4-W7	W4-W8
			-	-	-	0.01 (T)	0.05 (T)	0.20 (S)	0.28 (S)
Week 5	5.87 (1.45)		-	-	-	-	W5-W6	W5-W7	W5-W8
			-	-	-	-	0.03 (T)	0.19 (T)	0.25 (S)
Week 6	5.92 (1.24)		-	-	-	-	-	W6-W7	W6-W8
			-	-	-	-	-	0.17 (T)	0.23 (S)
Week 7	6.15 (1.46)		-	-	-	-	-	-	W7-W8
			-	-	-	-	-	-	0.05 (T)
Week 8	6.23 (1.45)		-	-	-	-	-	-	-
		**F $\eta_{p}^{2}$ ε**	**Week-by-Week Comparisons (Cohen’s d)**						
Feeling of being restored		F(7, 112) 10.05 *** 0.386 0.234	BL-W2	BL-W3	BL-W4	BL-W5	BL-W6	BL-W7	BL-W8
Baseline	4.88 (1.99)		0.29 (M)	0.69 (M)	0.95 (L)	0.73 (M)	0.78 (M)	0.81 (L)	0.84 (L)
Week 2	5.36 (1.34)		-	W2-W3	W2-W4	W2-W5	W2-W6	W2-W7	W2-W8
			-	0.51 (M)	0.82 (L)	0.56 (M)	0.63 (M)	0.66 (M)	0.70 (M)
Week 3	6.10 (1.53)		-	-	W3-W4	W3-W5	W3-W6	W3-W7	W3-W8
			-	-	0.29 (S)	0.06 (T)	0.04 (T)	0.14 (T)	0.18 (T)
Week 4	6.54 (1.51)		-	-	-	W4-W5	W4-W6	W4-W7	W4-W8
			-	-	-	0.22 (S)	0.29 (S)	0.15 (T)	0.10 (T)
Week 5	6.19 (1.61)		-	-	-	-	W5-W6	W5-W7	W5-W8
			-	-	-	-	0.02 (T)	0.08 (T)	0.13 (T)
Week 6	6.15 (1.17)		-	-	-	-	-	W6-W7	W6-W8
			-	-	-	-	-	0.12 (T)	0.17 (T)
Week 7	6.31 (1.55)		-	-	-	-	-	-	W7-W8
			-	-	-	-	-	-	0.04 (T)
Week 8	6.38 (1.60)		-	-	-	-	-	-	-

Notes: *** *p* < 0.001. T = trivial effect size; S = small effect size; M = medium effect size; L = large effect size.

## Data Availability

Data are available upon request for experts in the field.

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
