# Peer review of "Acceptance and Commitment Therapy (ACT) Improves Sleep Quality, Experiential Avoidance, and Emotion Regulation in Individuals with Insomnia—Results from a Randomized Interventional Study"

_life, 2021, doi:10.3390/life11020133_

Round 1

Reviewer 1 Report

This is an interesting study investigating the effectiveness of acceptance and commitment therapy for insomnia. I particularly like the investigation of how different factors (mostly experiential avoidance) might explain its effectiveness. However, I find the structure of the manuscript unclear, as the theoretical background as well as the design are not entirely clear at one point. I explain this in separate points below.

Introduction
- In the description of treatments for insomnia, I miss the recommendations made by the European and American Sleep societies. Both recommend CBT-I as first line treatment, but this is not clear from the introduction of available treatments.
- It is stated that there is an active control condition. This is not explained in the abstract. Also, it is unclear what the active control condition entails, besides meeting in the study center. What was done there? How is it an active control group?
- It is stated that a daily log is kept, allowing to “observe, if and at which time points ACT improved dimensions of sleep patterns and cognitive-emotional processes”. This sounds very interesting, but I think the authors should at least briefly state what was queried in the daily logs.
- Importantly: Experiential avoidance is stated to be a key concept to explain ACT-related improvements, but the concept is not introduced. As it is a key concept to ACT, and it is hypothesised that improvements in sleep quality will be associated with experiential avoidance, the concept should be clearly introduced and explained why it is a key concept to ACT.

Methods
- It is not at all clear how the participants are recruited (from what pool are these individuals recruited? Using what methods?).
- Only in the methods it becomes clear that ACT is given in group format.
- The flow chart is unclear: what happens at week 5? At week 5 it says “drop out by week 8” for the control condition. This is confusing.

Results
- It is problematic that there are so may baseline differences on important outcome variables (PSQI and SPAQ). This is barely mentioned and not taken into account in the other analyses. That can be a choice, but it should be at least acknowledged and discussed how this could have affected the results (also in the discussion).
- The micro analyses are interesting, but now show *many* comparisons. It is unclear to me why show this both in tables and in figures. Why not include figures only and show a separate line for treatment and control condition? That would make this much clearer to interpret.

Major overall concern:
- It is not entirely clear to me what the main focus of the authors is. Linking to the comments in the introduction: “experiential avoidance” is not clearly introduced, but it is the only measure for which the effects are shown in the results. This makes it difficult to follow the manuscript and interpret the results.
- In the introduction and discussion it reads as if the authors consider experiential avoidance as mediating the effects of ACT. However, this is not analysed as such.

Minor overall comments. The manuscript should be carefully reviewed and polished for inconsistencies, to name a few:
- The PSQI shows large baseline differences, but this is not mentioned in text.
- The outlining of the titles in Table 3 is inconsistent
- The labeling of the effect sizes in Table 4 is inconsistent: only [L] is presented in the table, but only [T], [S] and [M] are explained in the notes.

Author Response

Dear Reviewer,

Thank you very much for all your kind efforts.

We have addressed all concerns raised by the Reviewers. Please see the detailed point-by-point-response attached as a separate file. 

Reviewer 2 Report

Dear Editor,

Thank you for the invitation to review the manuscript entitled “Acceptance and Commitment Therapy (ACT) improves sleep quality, experiential avoidance and emotion regulation in individuals with insomnia – results from a randomized interventional study”.

The study aims are interesting and the writing clear for the most part. Below I have provided comments to consider:

METHODS

  1. Please describe the setting more in detail. How and from where were the participants recruited? Between which dates? Where they all included in the same intake?
  2. Was this study preregistered?
  3. Specify in the methods section at what timepoints the outcomes were measured (only mentioned in abstract)
  4. What is the amount of missing at follow-up? This is not presented in the flow chart. If you have more than 10% missing at follow-up: How are you dealing with that missing data? (Missing data analysis? Imputations?)
  5. Flow chart
    • The boxes have switched places (w5 versus w8 for the control group)
    • Why is the time point 5w reported?
    • Please report the time point 12 w
  6. How many were randomized? Abstract says 35, flow chart says 40.
  7. Please use the word ”measures” instead of ”tools”
  8. There are medium and large group differences in outcomes at baseline
    • This is a significant problem and should be adressed in the discussion. This implicates that the randomisation has been comprimised. How can this have occurred?
    • Please use ANCOVA instead of ANOVA to control for baseline differences
  9. Pearson correlations between change scores for changes occuring during the same timeframe can at best be an analysis that is hypothesis generating. Therefore the conclusion in the abstract that ”The pattern of results suggests that

ACT improved experiential avoidance, which in turn improved both sleep quality and sleep-related cognitive-emotional processes at longer-term in adults with insomnia.” needs to be tampered. As most of your change in sleep occurs within the first three weeks of treatment the change in experiential avoidance would need to have occured even earlier in order to establish an adequate timeline between outcome and presumed change mechanism. This limitation in the design should be mentioned in the discussion (given your second aim).

RESULTS

  1. The fact that your sample on average sleeps less than 4 hours at baseline is quite extraordinary even in an i-CBT trial! This makes me wonder if the participants are instead sleeping during daytime? Please compare this baseline value to those of other studies in the discussion.
  2. Regarding Table 4.
    • The title ”Descriptive and inferential statistical indices of physiological scores.” is misleading
    • The values of n2 should be in an own column as they now ”float in to” the F-values.
    • The abrevations ”[T] = trivial effect size; [S] = small effect size; [M] = medium effect size” are presented in the note but not consistently used in the Table.
    • The results on the Greenhouse-Geiser test show that the assumption of sphericity is not met in any of the ANOVAs. Thus, the F-value is likely to be inflated. Has this been corrected for?

DISCUSSION

  1. According to Table 2, 63% of the sample is female, in the discussion you say that 91% of the sample was female, what is correct?
  2. Please compare your results to those found in other trials of ACT for insomnia.

Author Response

(The authors gave the same response as above.)

Round 2

Reviewer 1 Report

Thank you for the clear overview of the revisions that were made. I think this greatly improved interpretation of the theoretical background and results.

Author Response

Again, we thank Reviewer #1 very much for their encouraging and positive comments and support throughout the whole review process.

Reviewer 2 Report

Dear Editor,

The Authors have improved the manuscripts in accordance with my comments. I have a few minor comments left, but I find no need to review the manuscript again:

  1. The Flow chart still needs editing. Please remove week 5 as a box in the Flow chart and instead report the timepoints BASELINE, POST and FOLLOW UP 
  2. On row 207-208 please replace the sentence 

    "Given this, to perform the statistics per protocol, the sample consisted of 35 participants." with a formulation like "All outcome analyses were performed on completer data."

  3. In the description of the statistical analyses the use of ANCOVA needs to be motivated. 

Kind regards

Author Response

Again, we thank Reviewer #2 for their encouraging and constructive comments and suppport throughout the whole review process. 
